# Biopolymers in Mucoadhesive Eye Drops for Treatment of Dry Eye and Allergic Conditions: Application and Perspectives

**DOI:** 10.3390/pharmaceutics15020470

**Published:** 2023-01-31

**Authors:** Anđelka Račić, Danina Krajišnik

**Affiliations:** 1Department of Pharmacy, University of Banja Luka-Faculty of Medicine, Save Mrkalja 14, 78000 Banja Luka, Bosnia and Herzegovina; 2Department of Pharmaceutical Technology and Cosmetology, University of Belgrade-Faculty of Pharmacy, Vojvode Stepe 450, 11221 Belgrade, Serbia

**Keywords:** ocular drug delivery, eye drops, mucoadhesion, biopolymers, penetration enhancers, bioavailability

## Abstract

Dry eye syndrome and allergic conjunctivitis are the most common inflammatory disorders of the eye surface. Although eye drops are the most usual prescribed dosage form, they are characterized by low ocular availability due to numerous barrier mechanisms of the eye. The use of biopolymers in liquid ophthalmic preparations has numerous advantages, such as increasing the viscosity of the tear film, exhibiting bioadhesive properties, and resisting the drainage system, leading to prolonged retention of the preparation at the site of application, and improvement of the therapeutic effect. Some mucoadhesive polymers are multifunctional excipients, so they act by different mechanisms on increasing the permeability of the cornea. Additionally, many hydrophilic biopolymers can also represent the active substances in artificial tear preparations, due to their lubrication and moisturizing effect. With the modification of conventional ophthalmic preparations, there is a need for development of new methods for their characterization. Numerous methods for the assessment of mucoadhesiveness have been suggested by the literature. This review gives an overview related to the development of mucoadhesive liquid ophthalmic formulations for the treatment of dry eye and allergic conditions.

## 1. Introduction

The eye is one of the most complex organs in the human body with a unique anatomy and physiology. Its characteristic anatomical structure and numerous physiological mechanisms enable protection of the eye against unfavorable and harmful external influences, but at the same time they represent the main problem for achieving and maintaining drug concentration in the target tissue, which should produce an appropriate therapeutic effect [1,2]. Local therapy, due to its simplicity of application, convenience, and non-invasiveness, is the most common route of drug administration in the treatment of diseases of the anterior segment of the eye. Eye drops make up about 90% of all ophthalmic products available on the market. However, the greatest challenge in the treatment of eye diseases is the short precorneal retention time of locally applied drugs on the surface of the eye and poor ocular availability, which is less than 5% [3,4]. To improve the therapeutic efficiency of existing drug formulations, various multifunctional biopolymers are being used to increase the viscosity of the tear film, exhibit bioadhesive properties, and/or to increase corneal permeability. Additionally, these substances are also used as active ingredients in artificial tear preparations due to their lubricating and moisturizing effect [5,6,7].

The objective of this paper is to give an overview of formulation approaches related to the introduction of mucoadhesive polymers, as functional excipients, into liquid ophthalmic preparations for the treatment of dry eye and/or allergic conditions. The review reveals the advantages of biopolymer application in viscous eye drops and the importance of appropriate evaluation of mucoadhesive properties, depending on the type and mechanism of interactions between polymers and mucin within the tear film.

## 2. Ocular Anatomy and Physiology

From the perspective of ocular drug delivery, the eye is divided into anterior and posterior segments. The anterior segment consists of the cornea, conjunctiva, iris, pupil, ciliary body, anterior chamber aqueous humor, trabecular meshwork, and lens, while the posterior segment consists of the vitreous humor, sclera, retina, choroid, macula, and optic nerve [8].

### 2.1. Cornea

The cornea is a colorless, transparent, nonvascularized and well-innervated layer of the anterior segment of the eye. Its main function is the refraction of light, which occurs due to its spherical shape, flat surface and the greatest refractive index of the ocular optical system [9,10]. The cornea protects the sensitive internal structures of the eye from external harmful influences, and it is the most sensitive portion of the eye, innervated by numerous free thin nerve endings (7000 pain receptors per mm^2^) [11]. The cornea consists of five to seven layers: surface epithelium, anterior limiting lamina (Bowman’s Layer), stroma (substantia propria), posterior limiting lamina (Descemet’s membrane), and inner endothelium (Figure 1). The epithelium is composed of a layer of basal cells which consists of four to six layers of squamous epithelial cells connected by tight junctions in order to form a barrier that prevents fluid loss and the penetration of foreign substances such as dust, bacteria, etc. [12,13].

The anterior boundary membrane (Bowman’s membrane) is an acellular connective tissue made of strong layered protein fibers, i.e., collagen. The stroma provides the largest part of the cornea (approximately 90%). It is mainly composed of water (78%), and the rest is dominated by collagen, which gives the cornea its strength, elasticity, and shape. The posterior basement membrane (Descemet’s membrane) of the corneal epithelium is a thin, but firm, layer of tissue, made of collagen fibers, which serves as a protective barrier against infections and injuries. The endothelium is a single-layered internal structure of the cornea, facing the anterior chamber. One of the most important functions of the endothelium is to keep the stroma clean and healthy with its strong intercellular connections, and to provide nutrients and molecules from the aqueous humor by an active transport mechanism [12,14].

### 2.2. Conjunctiva

Conjunctiva is the mucous membrane that lines the inside of the eyelids and the front of the sclera to the cornea. It is well vascularized, and the majority of the conjunctiva is made up of specialized stratified squamous cells. The goblet cells represent unicellular glands whose main function is to secrete mucin, which lubricates the eye and prevents the entry of microorganisms. These spherical or oval cells can secrete about 2.2 mL of mucus per day, which lubricates and protects the epithelial cells. The mucus reduces the surface activity of the tear film, thus enabling its stability [15].

### 2.3. Ocular Bariers

Several precorneal factors affect the appropriate delivery of topically applied drugs to the anterior segment of the eye. These include total dose/applied volume, nasolacrimal drainage, tear exchange, tear evaporation, tear dilution, tear pH, drug–protein interaction, drug metabolism, and the blood–retinal barrier [16]. The primary anatomical barriers (static barriers) for drug penetration into the eye are the cornea and conjunctiva. The epithelium and endothelium of the cornea, due to the presence of lipids, represent the basic barrier for the permeation of water-soluble substances, while the stroma limits the permeation of liposoluble substances. Therefore, for drug permeation through the corneal epithelium into the eye, it should have appropriate lipophilicity, but should also be hydrophilic enough to pass through the stroma [17]. Besides the cornea, the conjunctiva, as the first layer of the anterior segment, also represents a barrier for topically applied ophthalmic drugs. The sclera represents the next part of the eye barrier. Substances that permeate the blood vessels of the conjunctiva and the sclera pass into the systemic circulation.

The dynamic barriers of the eye include the physiological and reflex process of tear production, the blink reflex, and nasolacrimal drainage. In addition, tear fluid contains 2% or more proteins, some of which are enzymes such as esterases, aminopeptidases and monoamineoxidases, and lead to the inactivation of a large number of substances [18,19].

### 2.4. Tears and Lacrimal Drainage System

The tear film enables maintenance of the optical properties of the corneal surface, supplies oxygen and nutrients to the avascular cornea, lubricates the surface between the inner side of the eyelids and the corneal surface, removes foreign bodies and cells, and, with the help of lysozymes, ensures the antimicrobial properties of the cornea [20].

The tear film is formed by the activity of the lacrimal system, consisting of lacrimal glands (orbital and palpebral) and the secretory part (lacrimal sac, tear ducts). It is most often described as a complex mixture consisting of three main layers (Figure 2). The outermost is the lipid layer. It is secreted by the sebaceous glands (Meibomian glands) located on the edges of the eyelids. The lipid layer reduces the evaporation of tear fluid from the surface of the eye and ensures the stability of the tear film [20]. The middle aqueous layer, secreted by the lacrimal glands, consists of water, electrolytes, glucose, urea, and proteins. Change in the osmolality of the aqueous layer leads to a disorder of ocular surface homeostasis and can affect the retention and delivery of drugs administered on the ocular surface. The inner layer of the tear fluid is the mucous layer or mucin, which is produced by the goblet cells of the conjunctiva, but also by cells of the stratified squamous epithelium of the cornea and conjunctiva. It represents the thickest component of the tear film and gives it its viscosity [21].

The composition of tears varies depending on whether the eye is open or closed, stimulated or not, or in a pathological or physiological state [22]. The volume of tear fluid under physiological conditions is 7 to 10 µL, with an exchange rate of 16% per minute [20,23]. A part of the secreted tears is lost due to evaporation, and the rest passes through the nasolacrimal duct that opens into the lower nasal passage [20].

The tear film is only temporarily stable. Its stability can be disturbed in the presence of surfactants that dissolve the surface lipid layer, or caused by a decrease secretion of lipids, tears, or mucin. Disturbance of the tear film leads to an increase in the frequency of blinking, and is one of the frequent causes of dry eye syndrome [24].

### 2.5. Mucin and Mucoadhesion

The ability of polymers to enter into non-covalent interactions with mucin enables their adherence to the corneal and conjunctival surfaces of the eye. This property of the polymer enables longer retention of the applied liquid preparation on the surface of the eye, less loss due to drainage, better penetration of drugs, or longer local activity [5,7].

Mucus is an adhesive viscoelastic gel which covers most mucosal surfaces in the body. The main roles of the mucus layer are to maintain a healthy wet surface and to form a barrier against various pathogens, drugs, and other environmental toxic agents [25,26]. The mucus is composed from water (<95%, *w/w*), extracellular glycoproteins, lipids such as fatty acids, phospholipids, salts (~1%, *w/w*), carbohydrates, cholesterol, defensive proteins (i.e., lysosomes, defensins, trefoil factors, etc.) and mucin (<5%, *w/w*) [27,28]. Mucins are large extracellular glycoproteins with molecular weights from 0.2 to over 50 MDa that are negatively charged due to the presence of terminal sialic acid (pKa of 2.6) and sulphate groups [28].

Mucoadhesion is defined as the adhesion of a natural or synthetic polymer to a mucous membrane through physical or chemical interaction. The phenomenon of mucoadhesion is usually explained through two successive phases (Figure 3). In the first phase, contact is established between the mucoadhesive polymer and the biological substrate (in this phase, the ability of the polymer to spread over the biological substrate and interfacial forces play a key role); in the second phase, which can be defined as the phase of consolidation, there is physical interaction and the formation of secondary chemical bonds between the mucoadhesive polymer and the mucus located on the surface of the mucous membrane. In the case of mucoadhesive polymers, this phase is preceded by mutual interpenetration of polymer chains and mucin glycoproteins [29]. However, the exact mechanism of mucoadhesion depends on dosage form and is usually different in solid and liquid dosage forms. While the hydration process mainly affects the mucoadhesion of solid forms, the rheological characteristics of liquid forms are primary for their interaction with mucin [28].

Improving bioavailability using mucoadhesive drug delivery systems is a very useful approach applied in various routes of administration: oromucosal [30], nasal [31], ocular [32], and buccal [33].

After the local application of mucoadhesive eye drops, the polymer interacts with the glycoprotein, allowing formulation to spread over the surface. The therapeutically active substance is then able to diffuse into the target area. Formulations with mucoadhesive polymers have advantages such as prolonged residence time in the precorneal tissue, therefore they slow down the elimination of the drug, reduce the frequency of drug administration, and improve compliance. In addition, these polymers often have lubricating properties, which additionally affects the patient’s comfort [4,34]. On the other hand, the main disadvantage of this approach for the improvement of ocular availability is difficult dosing, which depends on the type of mucoadhesive polymer as well as the patient’s mucus production process [4].

### 2.6. Pharmacokinetics of Locally Applied Drugs

After local application to the lower conjunctival sac, the drug passes into the intraocular tissues through corneal and non-corneal pathways [35]. However, the bioavailability in aqueous humor is only 5% for lipophilic molecules and even below 0.5% for hydrophilic molecules [36]. Drug molecules cross the cell membrane to the greatest extent by the mechanisms of passive diffusion, facilitated diffusion, and active transport through carriers [37]. The cornea behaves like a typical cell membrane and solutes pass through this structure via transcellular and paracellular pathways [38]. At physiological pH, the surface of the cornea is negatively charged due to the carboxyl groups of proteins that build tight junctions. Therefore, the negatively charged molecules, due to repulsive forces, have difficulty in passing through the intercellular pores of the corneal epithelium, while positively charged molecules have better delivery to the cornea. The passive transport of molecules is influenced by various factors, such as lipophilicity, molecular weight, charge, and degree of ionization of the drug [3,38].

The volume of the precorneal tear film is about 7 µL, with the possibility of accommodating the conjunctival sac, whose capacity is approximately 15–30 µL [23]. The volume of one eye drop applied topically through commercial droppers ranges from 25 µL to 70 µL [39], but excess liquid is removed by drainage through the nasolacrimal duct and by blinking. Therefore, the application of more than one drop will not improve ocular availability, but may cause systemic toxic effects due to absorption. Additionally, tear secretion may be increased due to the stimulation of corneal sensory nerves, which are very sensitive to non-specific stimuli, such as pH, osmolality, drug properties, and formulation factors, leading to accelerated elimination of the drug from the precorneal space [40].

Most drugs that are applied locally to the eye are in the form of salts of acids and bases. Their molecules in the tear film are under the influence of the tear film pH value, depending on their concentration and properties. In addition to the pKa value of the drug, its molecular weight, the presence of preservatives and surfactants, composition of the vehicle, and the osmolality of the formulation are also factors that determine the pharmacokinetics of drugs in the eye [37]. Certain preservatives in topical formulations, such as benzalkonium chloride, can increase the ocular availability of drugs by causing changes in the cell membrane [41].

Based on all of the above, designing new drug delivery systems that can deliver the appropriate therapeutic concentrations to target tissues and maintain them without or with minimal side effects is the main focus of current research in the field of ophthalmic preparations [42,43].

## 3. Dry Eye Syndrome and Allergic Conjunctivitis

Dry eye syndrome and allergic conjunctivitis are the two most common inflammatory disorders of the eye anterior segment and are considered epidemics of the 21st century. Traditionally, dry eye syndrome and allergic conjunctivitis were considered two different diseases; however, recent literature data indicate the similarity and overlap of these two conditions due to the appearance of similar symptoms [44,45,46].

Dry eye syndrome is a multifactorial disease of tears and the surface of the eye, which leads to a feeling of discomfort, visual disturbances, and instability of the tear film with potential damage to the eye surface, often accompanied by an increase in the osmolality of the tear film and inflammation of the surface of the eye [47].

Based on new knowledge, the definition and classification subcommittee of the second Tear Film and Ocular Surface Society Dry Eye Workshop (TFOS DEWS II) revised the global dry eye definition [48]:

“*Dry eye is a multifactorial disease of the ocular surface characterized by a loss of homeostasis of the tear film, and accompanied by ocular symptoms, in which tear film instability and hyperosmolarity, ocular surface inflammation and damage, and neurosensory abnormalities play etiological roles*.”

Disturbed homeostasis of the tear film, including a quantitative and qualitative lack of tears, consequently leads to reduced moisture, increased friction, hyperosmolar stress, and chronic mechanical irritation of the eye surface [49]. The most common symptoms are a feeling of dryness, the presence of a foreign body, itching, pain, redness, and hyperemia [44,50]. The pathogenesis of dry eye syndrome is associated with cytotoxic inflammatory mediators, altered function of the lacrimal glands and nerves, metaplasia of the squamous epithelium of the conjunctiva, and a decrease in the number of goblet cells, all of which play a significant role in the damage to the corneal and conjunctival epithelium [51]. Without proper lubrication, there is significant damage to the cells on the eye surface, which leads to the exacerbation of the inflammatory process [52].

Currently, dry eye therapy usually involves the use of artificial tears and anti-inflammatory drugs [53]. Ocular lubricants represent the first line of treatment for dry eye syndrome and can prevent the appearance of a vicious inflammatory cycle [52,54]. Lubricants are substances that reduce friction during use, i.e., reduce the friction between the surface of the eye and the eyelids [55]. Over-the-counter (OTC) products known as artificial tears are largely used as a replacement or supplementation of the natural tear film [53]. The most common auxiliary substance in lubricating eye drops is water as a vehicle, with the addition of viscosity-increasing agents for the improvement of lubrication and prolonging the residence time on the eye surface [56]. Hydrophilic polymers, as viscosity-increasing agents, lead to an increase in the viscosity of the tear film, thereby stabilizing it, reducing tear loss, and protecting the tissue [57,58]. Furthermore, the potential mechanisms of action of ocular lubricants are addition to the volume of tears and reduction in the osmolarity of the tear fluid [58].

Mucin is a natural “lubricating” component of tears that gives viscosity to the tear film, so it is expected that preparations used in the treatment of dry eye will exhibit a mucomimetic effect. These preparations differ from each other in the choice of polymer or combination of polymers, concentration, length of action, viscosity, osmolality, and pH value. More viscous preparations can provide long-term relief; however, with extremely viscous ones, there is a greater possibility of blurred vision or polymer retention on the eyelids. The biggest drawback of the preparations used in the treatment of dry eye is still the short residence time at the site of application and the short-term effect of eliminating symptoms. Therefore, various approaches in the treatment of dry eye syndrome and new formulations (Table 1) using innovative polymers and their combinations are being intensively investigated [59,60,61].

Allergic conjunctivitis is a set of inflammatory eye processes mediated by mast cell activation that involve the conjunctiva, eyelids, and cornea. It often occurs together with other allergic diseases, including asthma, atopic dermatitis, and rhinitis. Eye allergies are primarily mediated by the activation and degranulation of mast cells because of the interaction of IgE with the allergen, with the involvement of T lymphocytes. Symptoms of an acute eye allergy include itching, redness, tearing, and swelling [44,62].

The basic therapy in the treatment of allergic conjunctivitis is oral and local antihistamines with the inclusion of topical corticosteroids for refractory forms. Ophthalmic NSAIDs, such as ketorolac, exert an antipruritic effect by reducing the level of prostaglandins in the tear film, but may lead to decreased compliance by causing a stinging sensation. The first generation of antihistamines, due to their anticholinergic effect, can cause symptoms of dry eye syndrome in patients with allergic conjunctivitis who are already predisposed to this condition, while novel mast cell stabilizers show better efficacy with fewer accompanying side effects [52]. Itching is a symptom used to differentiate allergic conjunctivitis from other forms of conjunctivitis. When symptoms of itching are clinically significant, there is a strong possibility that the eyes will be red and dry [63]. Hom et al. [44] revealed that patients with dry eye syndrome also have a higher chance of itching and erythema, which indicates a significant overlap of the symptoms of these two diseases, which makes differential diagnosis difficult. Allergic conjunctivitis has been found to cause changes in the composition of the tear film, which, as mentioned earlier, is associated with dry eye syndrome, or makes patients suffering from allergic conjunctivitis predisposed to dry eye syndrome. In addition, itching consequently leads to rubbing of the eye, which can damage the integrity of the corneal epithelium and worsen the inflammation on the surface of the eye and lead to pain [64]. In the chronic form of allergic conjunctivitis, there is frequent tearing of the tear film, a decrease in the content of conjunctival mucin as one of the layers of tear fluid, and a decrease in the number of goblet cells in the conjunctiva, which finally lead to dry eyes [44].

Therefore, dry eye syndrome and allergic conjunctivitis clinically have similar symptoms (itching, redness, burning, pain) which resemble each other (Figure 4).

**Table 1 pharmaceutics-15-00470-t001:** Studies on novel mucoadhesive formulations for dry eye syndrome treatment.

Type of Formulation/Drug Carrier	Active Substance(s)	Mucoadhesive Component	Assessment of Mucoadhesion	Reference
Nanoparticle	Epigallocatechin gallate	Hyaluronic acid	-	[65]
Nanoemulsion	Cyclosporine A	Chitosan	Mucoadhesive strength—texture analyzer, goat cornea	[66]
Nanoparticle	Cyclosporin A	Phenylboronic acid (PBA)	Assessment of covalent interaction between PBA and sialic acid using a spectrofluorometer	[67]
Nanostructured lipid carrier	Curcumin	Thiolated chitosan	-	[68]
Nanomicelles	Cyclosporine A	Hyaluronic acid	-	[69]
Niosomes	Tacrolimus	Hyaluronic acid	Surface plasmon resonance	[70]
Liposome	Hyaluronic acid, Crocin	Hyaluronic acid, Galactoxyloglucan	-	[71]
Nanoparticles	Tacrolimus	Gellan gum	-	[72]
Nanogels	Lysine-carbonized	Cationic lysine-carbonized nanogels	-	[73]
In situ gel	Vitamin B12	Pluronic F127 and hydroxypropyl methyl cellulose	In vitro assessment adhesion	[74]
Solution	Hyaluronic acid	Hyaluronic acid	Rheological method	[75]
In situ gel	Cyclosporine A	Chitosan	Ex vivo mucoadhesion study, bovine cornea	[76]
Solution	5-oxo-2-pyrrolidinecarboxylic acid	Hyaluronic acid	-	[77]
Nanogels	Poly(acrylic acid)/Polyvinylpyrrolidone	Poly(acrylic acid)	Rheological method; zeta potential measurements	[78]
Nanoemulsion	Ibuprofen	Chitosan	Rheological method	[61]
Cyclodextrin (CD)-based aggregate	Nepafenac	Sodium hyaluronate, sodium alginate	Mucoadhesive strength—texture analyzer, bovine cornea	[79]

## 4. Polymers as Functional Excipients for Improvement of Ocular Availability

Due to their low ocular availability, frequent dosing of high concentrations of drugs is often necessary to achieve an adequate therapeutic effect. Consequently, this can further lead to side effects and damage to cells on the surface of the eye, resulting in lower compliance [80]. Therefore, different formulation strategies are considered to increase drug concentration at the desired site of action and to improve ocular availability.

The use of mucoadhesive polymers that increase the viscosity of the tear film and penetration enhancers represent two approaches that achieve more pronounced therapeutic effects while reducing unwanted systemic effects, using lower concentrations of active substances. Ophthalmic preparations based on mucoadhesive substances and penetration enhancers are easy to manufacture and demonstrate excellent tolerance when applied to the cornea. The first approach leads to a significant improvement in the contact time of the preparation with the cornea, while the second approach implies an increase in trans-epithelial transport [81,82].

### 4.1. Mucoadhesive Biopolymers in the Ophthalmic Formulation

The first formulation approach to improve ocular availability refers to the introduction of polymers into the composition of liquid ophthalmic preparations. Hydrophilic polymers, primarily polysaccharides, and their derivatives are auxiliary substances in a certain number of liquid and hydrogel ophthalmic preparations, thereby increasing the local availability of applied drugs and improving the therapy of various eye diseases, acting through different mechanisms. The ability of the polymers to increase the viscosity of the tear film, as well as the manifestation of their bioadhesive properties, contributes to a significant reduction in drainage, prolonged retention of the preparation at the site of application, and better therapeutic efficiency of the applied drug [83]. In addition to viscosity, other factors, such as surface activity, adhesion to the eye surface, and, especially, interaction with mucin, are important for prolonging residence time on the eye surface [84]. Mucoadhesion is a more desirable polymer property than viscosity modification because low-viscosity solutions are better tolerated than viscous ones [5]. Generally, excessively viscous liquid preparations are poorly tolerated by patients due to the shearing forces that occur during eye movement and blinking. Therefore, pseudoplastic properties, i.e., reduction of viscosity due to an increase in shear rate during blinking (*shear-thinning*), are desirable in formulations intended for local ophthalmic application. Mucoadhesiveness and a pleasant feeling after application have significant roles for ocular availability, compliance, and final therapeutic effect [85].

For improvement of ocular availability, various hydrophilic polymers, such as polyvinyl alcohol [86], poloxamers [87], cellulose derivatives [88], hyaluronic acid [89], carbomers [90], and gellan gum [91], alone or in combinations, were used. All the listed polymers contribute to an increase in the viscosity of liquid ophthalmic preparations, although there is still no agreement on the optimal value of this parameter. The viscosity of most commercial eye drops ranges from 15 to 25 MPa·s. The viscosity of ophthalmic preparations in the range of 15–150 MPa·s enables longer retention of the preparation in the conjunctival sac and better resistance to the drainage system [19]. The viscosity of tear fluid is usually in the range of 1.3–5.9 MPa·s [92].

A large number of polymers within liquid ophthalmic formulations, including natural, synthetic, and semi-synthetic high-molecular substances, have the ability to form strong non-covalent bonds with mucin, i.e., have mucoadhesive properties [83]. Mucoadhesiveness is a property of some natural and synthetic macromolecules that allows them to remain at the site of application, which can achieve prolonged action. Some examples of mucoadhesive polymers for ophthalmic use are cellulose derivatives (methylcellulose, carboxymethylcellulose, hydroxypropylcellulose, and hydroxyethylcellulose), polyvinyl alcohol, polyacrylic acid, chitosan, and hyaluronic acid. Biodegradability, biocompatibility, and non-toxicity of natural biopolymers, especially glucosaminoglycans, make these substances excellent candidates for the development of modern ophthalmic preparations [83,85]. Hydration or degree of swelling; molecular weight; molecular conformation, flexibility and mobility of polymer chains; functional groups; and concentration are the most significant properties of polymers that influence the manifestation of mucoadhesiveness [93]. Molecular weight is generally one of the most significant factors affecting the functional characteristics of polymers, such as polysaccharides [94]. The basic characteristics of the most commonly used biopolymers in formulations for dry eye treatment are listed in Table 2. 

Hypromellose (HPMC) is most often included in the composition of artificial tears used in the treatment of dry eye syndrome. It has many –OH groups that form hydrogen bonds and hydrophobic methyl groups. Their amphiphilic nature provides the necessary hydrophobicity to bind to the hydrophobic epithelium and matrix glycocalyx, while providing a sufficient number of hydroxyl groups that bind water molecules, making them a substitute for mucin. In addition to the effect of cellulose ethers on the viscosity of the solution, these compounds have pronounced lubricating properties and increase the contact time of the preparation with the cornea due to their ability to form a film [37]. Although the literature often states that cellulose derivatives (HPMC, hydroxyethylcellulose, methylcellulose) possess mucoadhesive characteristics, the mucoadhesive capacity of nonionic derivatives is poorly expressed [100,101]. However, their mucoadhesive properties can be improved by covalent attaching of various ligands providing additional bonds to the mucus layer. The synthesis of betaine-modified HEC by adding N-chlorobetainyl chloride as a reagent and pyridine as a catalyst resulted in an increase in mucoadhesive properties [102].

Charged polymers, either anionic or cationic, express greater mucoadhesive capacity than nonionic polymers, such as cellulose ethers or polyvinyl alcohol. Cationic mucoadhesive polymers (e.g., chitosan) are taken into electrostatic interactions with negatively charged mucin, mostly because of the presence of amino functional groups that are protonated at the physiological pH [103]. Chitosan solutions have pseudoplastic and viscoelastic characteristics, which is desirable for liquid ophthalmic preparations, because of the tear film’s pseudoplastic character. The rheological properties of chitosan also enable retention of the preparation at the site of application, as the formulation is easily spread by blinking [103,104]. Chitosan effectively and non-selectively binds to mucosal surfaces in the biological environment [105]. The bioadhesiveness of chitosan was confirmed in an ex vivo study, in which radioisotope-labeled chitosan was applied in the form of a solution to a freshly isolated bovine cornea, and the activity of the radioisotope was measured with a scintillation counter. It is believed that the cationic nature and electrostatic interactions are responsible for the appearance of primary forces within mucoadhesion, which occurs due to the reaction of the positively charged amino groups of chitosan and negatively charged sialic acid residues present in mucin, the basic glycoprotein of mucus. Additionally, hydrogen bonding and hydrophobic interactions also play a role in the binding of chitosan molecules to mucosal surfaces. Furthermore, the pH of the chitosan solution, as well as the presence of other substances, contributes to this interaction. The property of mucoadhesiveness becomes more pronounced in the neutral and slightly alkaline environment that prevails in the tear film [104,106]. Application of chitosan for controlled drug delivery to the eye has been also intensively researched, since, in addition to mucoadhesive properties, significant biological characteristics such as antimicrobial activity and potential impact on wound healing can also play a role in the therapy of various eye diseases [107,108].

The interaction between negatively charged polymers (e.g., hyaluronic acid, sodium alginate, xanthan gum, pectin) and the corneal epithelium is primarily based on hydrogen bond formation and interpolymer diffusion [6]. Hyaluronic acid has excellent biocompatibility and mucoadhesiveness, as well as pseudoplastic and viscoelastic behavior. Hyaluronic acid can also be cross-linked, so it is used as an excipient in the composition of films, inserts, and nanoparticles with the possibility of achieving prolonged retention, release, and action of incorporated drugs. Compared to chitosan, hyaluronic acid has less pronounced mucoadhesive properties, i.e., in order to achieve a similar level of mucoadhesiveness, it would be necessary to use hyaluronic acid in a concentration that gives extremely viscous solutions, which lead to a feeling of discomfort in patients and undesirable reflex blinking [85]. Hyaluronic acid molecules have similar physical characteristics and composition to tear glycoproteins and thus easily coat the corneal epithelium. Polymers adsorbed on the mucin–water layer interface extend into the adjacent water phase, which stabilizes the thicker water layer [7]. Numerous studies have shown a significant improvement in the ocular availability and therapeutic effect of formulations containing hyaluronic acid compared to commercially available preparations with the same drug [69,109,110].

It is known that hydrophilic bioadhesive polymers possess a large number of functional groups, such as sulfate, hydroxyl, carboxyl, and amide, which form numerous non-covalent bonds with the mucin layer in the precorneal area and on the conjunctiva wherever mucin is present [100]. In ophthalmic preparations, bioadhesive non-hydrated polymers are used in concentrations where they exhibit optimal viscosity. In the swollen, i.e., hydrated, state, the distance between the polymer chains leads to the flexibility of the polymer, which enables uniform spreading on the cornea and the conjunctiva. The polymer–mucin interaction force occurs when the mucoadhesive polymer interacts with the corneal epithelium. Using in vitro viscometric data, the aforementioned interaction can be defined as the polymer–mucin interaction force in the corneal region [37].

Wash-out of mucoadhesive polymers depends more on the rate of turnover of the mucus layer (approximately 15 to 20 h) than on the turnover of the tear fluid [111,112]. In addition to the ability to prolong the retention time of the drug in the precorneal surface, mucoadhesive polymers can be capable of protecting the eye surface from the irritating effect of some drugs [37]. They also appear as active substances in artificial tears used in the treatment of dry eye due to their lubricating and moisturizing effect on the eye. Polymers used to increase viscosity also have certain disadvantages, such as causing blurred vision, damaging the ocular epithelia, and resulting in possible patient discomfort [4]. These side effects are mainly due to higher viscosity; therefore, it is necessary to use appropriate polymer concentrations. Too high a viscosity of liquid ophthalmic preparations can also be undesirable for production, as it can make the filtration process difficult, especially during sterilization (the pore size of membrane filters is usually 0.22 µm). Additionally, high viscosity can lead to problems with dosing due to difficulty in determining an accurate dose of the drug [4,57]. The choice of bioadhesive polymer for the ophthalmic formulations depends on numerous factors, but above all on the viscosity and wetting properties of the polymer. The viscosity of the polymer depends on the molecular weight, concentration, temperature, and shear force. Polymers that exhibit non-Newtonian behavior affect the pseudoplastic properties of a formulation. The viscosity of these formulations decreases with an increase in shear rate, which results in a significantly lower blinking resistance and better acceptance by patients [58,103].

During the formulation of ophthalmic preparations, combinations of polymers that exhibit a synergistic effect on viscosity and rheological behavior or enable more pronounced mucoadhesive properties can be used. It has been confirmed that the combination of hyaluronic acid and chitosan, and also hyaluronic acid and polysaccharides from tamarind seeds (lat. *Tamarindus indica*), exhibits a synergistic mucoadhesive effect [113,114]. The presence of gelatin improves the mucoadhesive properties of carrageenan [115] and hyaluronic acid, due to the formation of a soluble polymer–polymer interaction product [116]. The combination of trehalose and hyaluronic acid reduces the desiccation of membrane lipids and proteins and improves the viscoelasticity of the tear film [117]. In an in vitro study on cultured human corneal epithelial cells, the positive effect of combining hyaluronic acid and hydroxypropyl guar gum was confirmed, when compared to the application of these polymers individually. The combination of polymers enabled more efficient hydration and lubrication, i.e., reduction of drying over a longer period of time, compared to the individual polymers [118]. There are also confirmations in the literature about the positive rheological synergism of xanthan gum and guar gum, which through mutual interaction contribute to a longer retention of the preparation at the site of application and better ocular availability [119].

Therefore, consideration of the physicochemical and functional characteristics of polymers is necessary in the early formulation phase of mucoadhesive ophthalmic preparations. These data are required to display the desired action with minimal side effects and irritating potential [5,6]. Some advantages and disadvantages of biopolymers in ophthalmic preparations are shown in Figure 5.

### 4.2. Use of Penetration Enhancers

Penetration-enhancing substances could be added to ophthalmic preparations to modify the permeability of the corneal epithelium. The basic categorization of penetration enhancers is: (a) chelating agents (citric acid, EDTA), (b) surfactants (sodium lauryl sulfate, cetylpyridinium chloride, benzalkonium chloride, Tween 20, parahydroxybenzoic acid esters), (c) cyclodextrins, (d) bile salts (sodium glycolate), and (e) fatty acids (oleic and caprylic acid) [120]. Penetration enhancers improve the ocular availability through various mechanisms that lead to a reduction of the barrier function of the eye mucous membranes by temporarily changing the structure or properties of these membranes. The main three mechanisms of penetration enhancers are modifying the stability of the tear film and mucus layer, increasing the permeability of cell membranes by membrane fluidization, and reversible opening of tight junctions. Most of these substances act simultaneously through several mechanisms [120,121].

There are numerous data which prove that the cationic polymer chitosan is actually the polymer of choice for ophthalmic application because, in addition to its mucoadhesive character, it also acts as an enhancer of the permeability of cell membranes by influencing both the intercellular and intracellular pathways of epithelial cells, without causing damage to the cell membrane and affecting viability [104]. Chitosan improves the penetration of active substances through the paracellular route by opening tight junctions between epithelial cells. In in vitro studies, chitosan has been shown to decrease transepithelial electrical resistance (TEER), which indicates a weakening or opening of tight intercellular junctions [122]. In addition, literature data also indicate the role of chitosan in increasing cell permeability through various additional mechanisms. Namely, it was announced that chitosan leads to local distortion of phospholipid chains, caused by a combination of hydrophobic, dipole, and electrostatic interactions [123]. Chitosan has also been shown to increase the cell membrane permeability of bacteria due to the electrostatic interaction between its NH^3+^ groups and phosphoryl groups of cell membranes’ phospholipid components [124].

Most chemical penetration enhancers, especially surfactants and chelating agents, can potentially cause unwanted and toxic effects after long-term use. As a result of their accumulation in the cornea, lesions with infections develop, as well as mechanical damage and possible irritation. Therefore, it is necessary to use the minimal concentrations required for the expected effect without causing irritation, discomfort, or damage [125].

Local application of formulations that contain mucoadhesive polymers in addition to penetration enhancers enables improved efficacy of the drug in lower concentrations and prolonged delivery with reduced toxicity and minimal side effects. Mucoadhesive polymers, such as chitosan, hyaluronic acid, and alginate, enable optimal activity of the penetration enhancer, while at the same time mitigating damage to the integrity of the epithelium and reducing the possibility of irritation [120].

## 5. Characterization of Mucoadhesive Ophthalmic Preparations

Modifications of conventional pharmaceutical dosage forms and the development of modern carriers require the application of new methods of analysis to achieve the most complete pharmaceutical–technological and biopharmaceutical characterization. Depending on the pharmaceutical dosage form, additional tests, such as encapsulation efficiency testing, interactions between the active substance and the carrier, the possibility of gel formation with in situ gelling systems, mucoadhesivity, in vitro/ex vivo transcorneal permeation, and cytotoxicity/biocompatibility tests can also be carried out [126].

As mentioned before, the addition of viscosity-adjusting agents to eye drop formulations can significantly improve the ocular availability of the drug. By testing the viscosity of eye drops, it is possible to determine the appropriate polymer concentrations in order to achieve an optimal effect on the retention of the formulation and avoid unwanted effects on the surface of the eye, such as blurred vision or blinking with difficulty. Viscous solutions obtained in this way usually have pseudoplastic, i.e., time-dependent, properties (*shear thinning*), and sometimes it is difficult to distinguish between highly viscous solutions and gels. For this reason, it is important to test the flow type of eye drop formulations modified in this way. Viscous solutions containing hydrophilic polymers are usually liquids that exhibit a non-Newtonian type of flow. Nevertheless, from the perspective of increasing ocular availability, the capacity of polymers to adhere to biological membranes is more important than their ability to lead to an increase in viscosity, because liquid preparations of lower viscosity are better tolerated by the eye [5,7].

### 5.1. Assessment of Mucoadhesive Properties

Over the years, various in vitro and in vivo methods have been used to evaluate mucoadhesive characteristics [127,128,129]. One simple and easily performed method is the turbidimetric one. This method is based on measuring the turbidity of the mucin dispersion (0.1%, *w*/*v*) after incubation with the tested sample. After mixing the polymer solution with the mucin dispersion, in case of interaction, precipitates appear and the turbidity changes in relation to the mucin dispersion turbidity. The turbidity of the mixture (1:1) is continuously monitored spectrophotometrically at 650 nm (wavelength of visible light) for 6–8 h, and a comparison is made with the turbidity of the mucin dispersion [101,130]. In a mixture of mucin dispersion and a solution containing a mucoadhesive polymer, e.g., chitosan, turbidity increases over time [101]. The increase in turbidity indicates the mutual interaction of chitosan and mucin. Namely, positively charged amino groups of chitosan, at a neutral pH value, react with negatively uncharged sialic acid residues present in the composition of mucin, resulting in the occurrence of precipitation, which is reflected in an increase in turbidity [130,131]. In contrast, in mixtures of solutions containing other polymers (e.g., hyaluronate or hydroxypropyl guar gum) and mucin dispersions, a decrease in turbidity occurs due to the absence of mutual interaction of the polymer with mucin and constant mixing over time. Based on such results, it could be concluded that such polymers do not show affinity towards mucin [101]. However, literature data indicate a well-expressed mucoadhesive character in hyaluronate [89]. Such results only indicate the importance of choosing an appropriate method for the evaluation of mucoadhesive characteristics depending on the type of interaction between polymer and mucin [127].

The measuring of zeta potential, which is also called the mucin particle method, is a useful technique for revealing the mucoadhesive characteristics of polymers that enter electrostatic interactions with mucin. It is based on the change in the zeta potential of the mucin dispersion after incubation with a polymer that has an affinity to interact with mucin [101,131,132]. Binding of the polymer causes changes in the surface properties of the mucin particles, which results in changes in the zeta potential [113]. The value of the zeta potential of the mucin dispersion is ~ −8 mV [101,132]. The negative charge of mucin particles can be explained by the presence of sialic acid residues, i.e., oligosaccharide chains that contain a large number of carboxyl and sulfate groups [96]. After mixing the mucin dispersion with chitosan-containing solutions, the zeta potential changed from negative to positive values (21 mV) due to the presence of protonated amino groups of chitosan [101,133]. On the other hand, when adding the solution with hyaluronate to the mucin dispersion, the negative value of the zeta potential increases significantly (−25 mV), and this increase in zeta potential is explained by the anionic nature of hyaluronate and the presence of a large number of carboxyl groups in the structure of this polymer. The method based on the change of the zeta potential is adequate for application to samples containing positively charged polymers, such as chitosan [101].

One of the most commonly used methods for evaluation of the mucoadhesive characteristics of polymers is based on determination of the polymer–mucin interaction by measuring the change in viscosity within the framework of continuous rheological analysis. A rheological approach to the assessment of the ability of a polymer to adhere to mucous membranes is based on measuring the increase in viscosity or viscoelasticity after mixing it with a mucin dispersion [134]. Mucoadhesion force is estimated based on changes in viscosity in a system where interaction energy can be transformed into activity, which results in redistribution of macromolecules and changes in viscosity. Therefore, mucoadhesiveness is detected when the rheological response of the polymer–mucin mixture is greater than the contribution of the polymer solution and the mucin dispersion individually. In addition to continuous rheological analysis, an oscillatory rheological analysis can be carried out simultaneously to analyze the elastic behavior of the sample in addition to the viscosity. The change in viscosity occurs as a result of a synergistic effect, while viscoelastic oscillatory measurements can indicate the type of mucoadhesive interaction [127,135]. The viscometric method of assessing mucoadhesive properties is based on measuring the viscosity in the following samples:Sample (a): mucin dispersion 10% (m/m);Sample (b): tested polymeric solution;Sample (c): mixture of mucin dispersion 10% (m/m) and tested solution.

Based on the measured viscosities, the “rheological synergism” parameter (syn. Mucoadhesive index) is calculated based on the following equation [134]:∆η = ηc − (ηb + ηa)(1)

If the tested solution contains a mucoadhesive polymer, the viscosity of the mucin dispersion mixture and the solution (ηc) has a higher value than the sum of the individual viscosities (ηa + ηb) due to the interactions that occur between the polymer and the mucin, and in that case a positive mucoadhesive index value is obtained [134,136]. When the mucoadhesive index increases with shear rate, it indicates that the mutual interaction between polymer and mucin is more pronounced at higher shear rates. This phenomenon can also be explained by the more pronounced pseudoplastic character of polymeric solutions compared to mixtures with mucin dispersion.

The pseudoplastic behavior of ophthalmic solutions is considered highly desirable as it provides adhesion to the mucosal surface even at the high shear rates that occur during blinking [137]. On the other hand, if the value of the mucoadhesive index is decreasing with an increase in shear rate, it can be concluded that the interaction with mucin is more pronounced in the periods of rest between blinks [89,96]. The application of the results of rheological characterization for the assessment of mucoadhesiveness has certain limitations, because the results obtained in this way are influenced by the polymer concentration, the viscosity of the polymer solution, the type of mucin and its concentration, the pH value, and the type of medium [137]. Conditions such as polymer concentration, type and concentration of mucin, pH, and type of medium can be constant, except for the viscosity of the solutions containing different polymers [101]. To compare the mucoadhesive properties of polymeric solutions with different functional characteristics, i.e., to eliminate the influence of the viscosity of different polymers, the “normalized rheological synergism” parameter (syn. normalized mucoadhesive index) is also calculated. The value of this parameter is obtained by dividing the value of ∆η by the apparent viscosity of the tested solutions [101,137]. In addition, wettability and spreadability are thought to play a major role in binding to mucin, rather than mutual interpenetration of the polymer and glycoprotein chains, so that other techniques, such as contact angle measurement, are even more predictive of in vivo conditions [135].

Interaction in the polymer–mucin mixture can be evaluated by oscillatory rheological analysis. The mechanisms of polymer–mucin interactions can be based on physical entanglement, ionic interaction, van der Waals forces, and hydrogen bond formation [7]. The relative magnitudes of the elastic (G′) and viscous (G″) moduli can indicate the qualitative characteristics of the structures in the mixture. In solutions containing high-molecular-weight polymers, there are three possibilities: G′ >> G″ for chemically cross-linked systems, G′ > G″ for systems connected by secondary bonds, and G′ ≤ G″ for physically cross-linked polymer systems [90]. In a study related to the assessment of interactions in polymer-solution–mucin mixtures, oscillatory measurements were performed. Three polymer solutions with chitosan, alone (F2M) and in combination with hydroxypropyl guar gum (F7M), i.e., with hydroxyethylcellulose (F8M), were tested. The obtained results (Figure 6) indicated the physical intertwining of polymer chains and mucin glycoprotein chains as well as the creation of weak hydrogen bonds between the crossed chains, because G′ ≤ G″, in all test samples. The crossing of G′ and G″ in the region of lower frequencies confirmed the structural behavior within the mixture [101]. The parameter calculated from the ratio of viscous (G″) and elastic (G′) moduli is called the loss factor or the tangent of the phase angle (tan δ). Viscoelasticity is the property of a material that exhibits both viscous and elastic properties under the influence of deformation. Viscoelastic liquids have a phase angle equal to or greater than 90° and exhibit a higher value of the viscous (G″) than the elastic (G′) modulus (tan δ > 1). In contrast, gels, as solid viscoelastic materials, have higher values of the elastic (G′) than the viscous (G″) modulus (tan δ < 1) and exhibit a time-dependent, delayed response, either during the application of stress and deformation or after their termination [138]. The tan δ value indicates a dominant elastic or viscous behavior in the polymer–mucin mixture, which affects the efficiency of their interaction [101].

### 5.2. Consideration of Surface Tension

In addition to the numerous quality requirements that eye drops must meet to be physiologically acceptable (pH, tonicity, viscosity), far less attention has been paid to surface tension adjustment. Determination of the surface tension of eye drops is a very useful technique to assess the compatibility of eye drops and tear fluid [139]. The surface tension of tear fluid depends on the presence of mucin, lipocalin, and lipids. The normal value of the surface tension at the air–tear fluid interface, which ensures the appropriate stability of the tear film, is in the range of 40–46 mN/m and ophthalmic preparations with a surface tension of less than 35 mN/m on the eye cause feelings of pain and discomfort. As a result, there is increased secretion of tears and more frequent blinking, which all leads to faster washing of the drug from the eye surface and a decrease of ocular availability [139]. In general, the lipid layer of the tear film is the key factor responsible for the surface tension of the tear fluid and its ability to maintain the surface tension is important for the stability of the tear film [140].

Polymers in eye drops increase their viscosity, while simultaneously reducing surface tension. Viscous solutions with a lower surface tension spread more easily on the eye surface and adhere better to the surface of the cornea [57]. Moreover, surface tension can affect the production process, as well as the accuracy of dosing, i.e., the uniformity of the volume of drops from multi-dose containers [141]. Literature data show that HPMC exhibits some surface activity, while hydroxyethylcellulose shows a very moderate decrease in surface tension. When using substances that do not affect surface tension, such as hydroxyethylcellulose, consideration should be given to adjusting the surface tension to the acceptable limits of the physiological range [139]. In the treatment of dry eye syndrome, it is desirable that the eye drops have a slightly lower surface tension than the physiological values of the surface tension of the tear fluid, because this condition is associated with higher values of surface tension at the air–tear fluid interface, which can range from 44–53 mN/m [142]. In addition, eye drops with a lower surface tension have significantly better eye-wetting properties, mix easily with the ingredients of the tear film, and spread well over the surface of the cornea, which results in a longer stay in the precorneal area and thus a longer and more pronounced therapeutic effect [143,144].

### 5.3. In Vitro and In Vivo Models Developed for Biopharmaceutical, Biocopmatibility, and Efficiency Assessment of Polymeric Eye Drops

In the past, biopharmaceutical characterization of ophthalmic preparations was largely based on the use of animal models, while, in the last few decades, significant progress has been made in the development of new techniques based on the application of ex vivo tissue, in vitro cell models, and in silico methods [145].

Evaluation of drug permeability through the eye barriers is one of the most important steps in the assessment of ocular availability and/or bioequivalence. Auxiliary substances included in the formulation can significantly affect the permeation of drugs through the eye barriers, and, therefore, changes in the formulation can significantly affect the permeation of the active substance. Cell and tissue models are equivalent models for the evaluation of ocular availability. They are not routinely used in industry and are still not implemented in standard regulatory protocols due to the lack of validation of such methods. In the last decade, intensive work has been done on the development of models of the anterior segment of the eye (especially models of the cornea) [146,147,148].

Various models are used to predict corneal permeability, most often those based on ex vivo methods and the use of vertebrates and in vitro cellular methods. To reduce the number of sacrificed laboratory animals to a minimum and avoid problems related to ethical issues, in vitro cell methods are increasingly used. Cell-based models most commonly involve the use of primary cell cultures, immortalized cell lines, or reconstituted tissue cultures of rabbit or human origin [149].

Furthermore, numerous in vitro transcorneal permeability models have been developed that differ in the type and origin of the cells (animal/human), the material from which the semi-permeable membranes are made, the cell culture protocol, the composition of the culture medium, and the length of time the cells are cultivated and exposed to air [146,147,150,151]. An immortal cell line based on human corneal epithelial cells (HCE-T) has been characterized to the greatest extent in terms of the porosity and pore size of the formed epithelial barrier in vitro and passive paracellular and transcellular transport [146,152,153]. In addition to simplicity, reproducibility, and the possibility of testing a large number of samples, the application of modern techniques enables the reduction of costs and the time required for conducting experiments. Such in vitro models were used for assessment of the effect of chitosan on olopatadine permeation through two corneal HCE-T cell-based models: Model I (poor barrier function and low TEER values) and Model II (improved barrier function and high TEER values). Higher values of the permeation coefficient (P*_app_*) for the formulations containing chitosan were observed both on Model I (35.1 × 10^−7^ cm/s) and on Model II (5.5 × 10^−7^ cm/s). These results represent another confirmation that the action of chitosan as a penetration enhancer is based on different mechanisms for the reversible disruption of tight junctions between epithelial cells, such as the interaction with cell membrane phospholipids, as mentioned above [154].

One of the modern methods of predicting the corneal permeability of active pharmaceutical ingredients from eye drops is based on the application of parallel artificial-membrane permeability assay (PAMPA). The PAMPA technique can be considered a useful alternative in the early stages of the development of ophthalmic preparations, compared to expensive and time-consuming ex vivo and in vitro cell methods. Using this technique, transcellular permeability is determined because the artificial membrane is based on a phospholipid layer [145]. Considering that the epithelial layer of the cornea is responsible for 99% of resistance to the diffusion of drugs, while the other layers (stroma and endothelial layer) can partially influence the diffusion of highly lipophilic compounds [155,156], it has been concluded that PAMPA represents a useful technique for the prediction of corneal permeability for active substances for which the hydrophilic stroma will not be a limiting factor [145,154]. In the study [154], the PAMPA technique was also used for evaluation of the impact of polysaccharide on the permeation of olopatadine. Although this artificial membrane was based only on phospholipids, the obtained results for P*_app_* were correlated with the results obtained on in vitro cell-based models, when the influence of chitosan on the value of this parameter was observed.

In vivo tests of ophthalmic preparations, besides the determination of bioavailability and bioequivalence, are also performed to assess ocular tolerability. For in vivo tests of ophthalmic preparations, in most studies, rabbits and other animal species are used as experimental animals, e.g., dogs [157]. Due to its controversial nature, the use of the Draize test in the USA and Europe has decreased in recent years, and it is sometimes modified using anesthetics and lower concentrations of test substances. Substances for which the occurrence of side effects has already been determined in vitro are not used when performing the Draize test, which leads to a reduction in the number of tests [158]. Wherever possible, alternative experimental in vitro models (e.g., cell cultures, isolated organs, microorganisms) and computer simulations should be used instead of in vivo experiments. The request to reduce the use of experimental animals refers primarily to toxicological tests. Where the legislation of the European Union does not recognize any alternative method, it is necessary to use the smallest possible number of experimental animals adhering to statistical evaluations [159].

In vitro cultures of human corneal epithelial cells and the tetrazolium salt (MTT) reduction test are useful techniques for conducting preliminary cytotoxicity studies of new ophthalmic formulations. As mentioned before, 3D models of human corneal cells have been proposed as useful alternatives to the classic Draize test in order to assess potential irritant effects on the eye, following the application of pharmaceutical and cosmetic products [160]. The MTT test has been shown to be a fast and cost-effective method for screening the cytotoxicity of new formulations, and the procedure performed on 3D corneal models provides a good correlation between in vitro results and modified maximum average score (MMAS) values that correspond to maximum average score (MAC) obtained after 24 h or more in an in vivo experiment. The use of immortal cell lines is extremely acceptable in terms of availability, reproducibility, ease of maintenance, and ease of damage detection [157,160].

The investigation of new ophthalmic formulations, including those containing different polymers, necessarily involves biocompatibility testing. This testing primary includes cytotoxicity or cell proliferation in vitro tests [161]. In a recent study, biocompatibility assessment of different formulations of eye drops with olopatadine (0.1%, *w*/*v*) and polysaccharide polymers (chitosan—0.5%, *w*/*v* and hydroxypropyl guar gum—0.25%, *w*/*v*) was performed. The viability of corneal HCE-T cells was evaluated by performing the MTT test procedure. All formulations containing polymers, alone or in combination, showed satisfactorily cell viability, over 80%. Moreover, cell viability was higher after incubation with formulations of the most complex composition, i.e., olopatadine formulations that contained combinations of two polymers (96.9 ± 3.4%) compared to formulations with single polymers (90.5 ± 4.5% for chitosan, i.e., 85.9 ± 5.3 for hydroxypropyl guar gum). The combinations of polymers showed lower cytotoxic potential, i.e., a complete absence of cytotoxicity, compared to each polymer individually [154]. The satisfactory low cytotoxic effect of the polysaccharide polymers used agreed with previously published results. In vitro (corneal HCE cells and conjunctival NHC-IOBA cells) and in vivo studies examining different formulations using chitosan as a mucoadhesive polymer showed no irritation or visible adverse changes in corneal or conjunctival epithelium morphology [104,162]. Additionally, satisfactory cell viability was obtained after treating a corneal epithelial model based on the CEPI 17 cell line with artificial tears containing hydroxypropyl guar gum [99].

To evaluate the pathophysiological aspects and monitor the effectiveness of ophthalmic preparations for the treatment of allergic conjunctivitis, various animal models have been developed. In vivo studies involve the administration of allergens (pruritogens) to an experimental animal, which leads to a characteristic behavior that can be quantified [163,164,165].

Itching is a characteristic symptom of various forms of conjunctivitis, which in animal subjects is manifested in the form of characteristic scratching of the eye with the hind paw. Scratching the eye can lead to damage to the surface of the eye, causing additional inflammation and increased itching. As a result of damage to the integrity of the cornea, which is extremely sensitive to algogenic stimuli, pain may occur. Therefore, it is significantly effective to suppress the feeling of itching in order to prevent the occurrence of additional damage and the appearance of pain [166]. Rodents, guinea pigs [167,168], and mice [169,170] were most often used as animal subjects in preclinical models of allergic conjunctivitis. Mice of the C57BL/6 strain are recommended as the most suitable animal subjects for establishing allergic conjunctivitis models and testing potential new treatments [154,166,171]. The efficacy of an olopatadine eye drop formulation containing a combination of chitosan and hydroxypropyl guar gum as viscosity increasing agents was evaluated in a study using an eye itch test, in which histamine as a pruritogen was instilled into the lower conjunctival sac of mice. The specific behavioral response that occurred was quantified as 17.1 ± 1.2 scratching and 10.7 ± 0.9 wiping bouts [154]. A set of rapid hindpaw movements in response to instillation of histamine is thought to be an itch sensation in rodents and would be a paradigm for histamine release from conjunctival immune cells in human allergic conjunctivitis. Itching that occurs in the eye is a pathognomonic symptom of allergic conjunctivitis, but the sensation of pain can also be present or caused by strong scratching of the cornea [166]. Nociceptive behavior is manifested in rodents by gentle rubbing movements of the eye with the ipsilateral forepaw, but it is less pronounced compared to the behavior that reflects the itch sensation [172]. The polymeric olopatadine formulation has shown slightly more pronounced anti-itch/analgesic effect than the commercial preparation, which did not contain polysaccharide polymers [154].

## 6. Concluding Remarks

Eye drops are the most common dosage form of ophthalmic drugs. However, the greatest challenge in the treatment of eye diseases is maintaining the therapeutic concentration of the drug at the site of application due to the presence of numerous barrier functions of the eye. By adding hydrophilic polymers that possess mucoadhesive properties, it is possible to increase the viscosity of the tear film, reduce drainage, and contribute to prolonged retention of the preparation at the site of application and better ocular availability of the drug. In addition, a large number of hydrophilic biopolymers possess lubricating and moisturizing properties, which at the same time exhibit a favorable therapeutic effect in the treatment of conditions such as dry eye syndrome and various allergic conditions. Some polymers, such as chitosan, apart from their favorable mucoadhesive characteristics and influence on viscosity, also improve drug permeation by acting through different mechanisms. The combination of two or more polymers significantly contributes to viscosity and lubrication, and allows decreasing the concentration of individual polymers, reducing the possibility of side effects. In the future, it will be necessary to standardize methods for evaluation of the functional characteristics of polymers, which can be crucial in the achievement of therapeutic effects in various eye disorders.

## Figures and Tables

**Figure 1 pharmaceutics-15-00470-f001:**
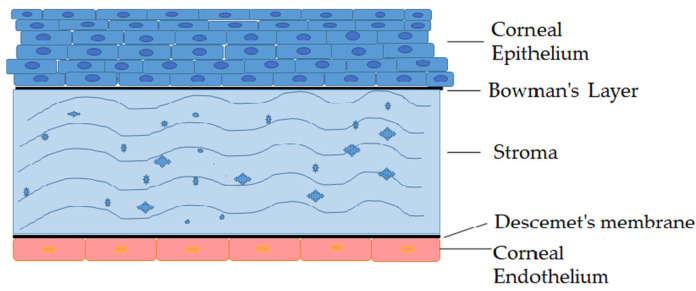
Schematic of human cornea structure.

**Figure 2 pharmaceutics-15-00470-f002:**
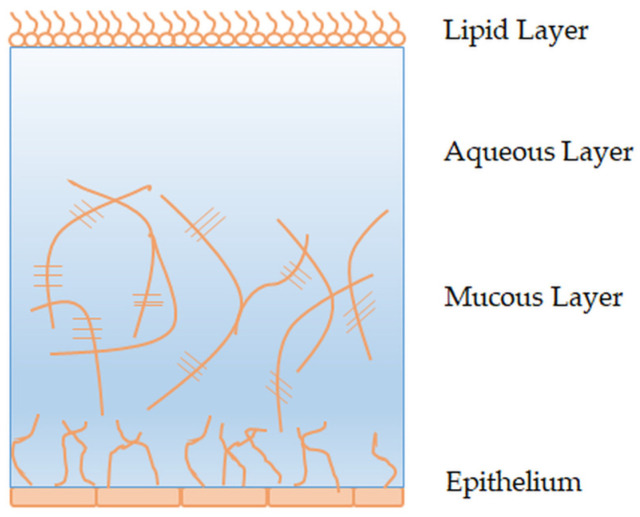
Tear film structure.

**Figure 3 pharmaceutics-15-00470-f003:**
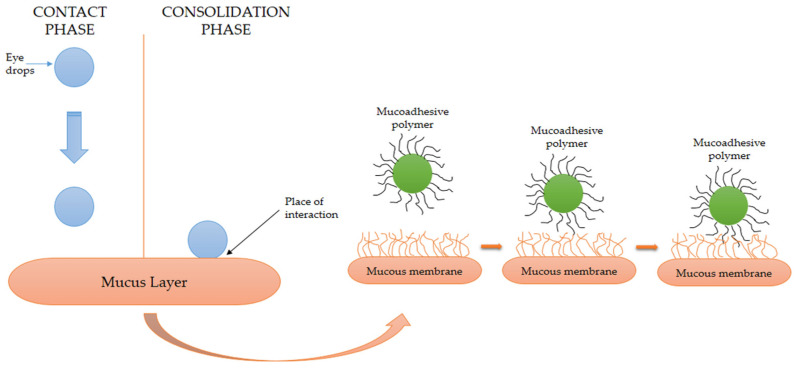
The mechanism of mucoadhesion at the ocular surface.

**Figure 4 pharmaceutics-15-00470-f004:**
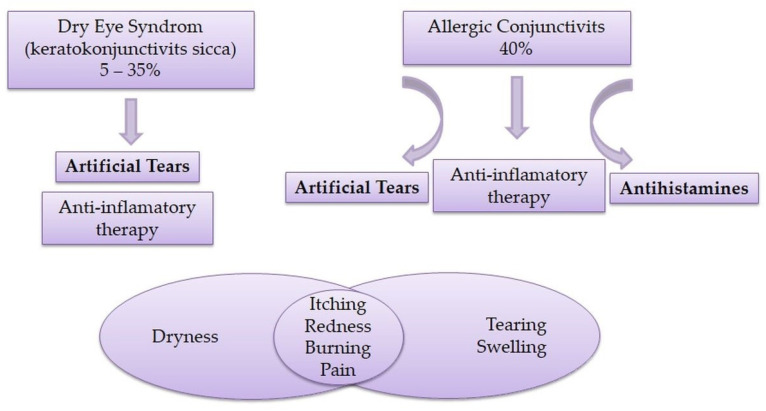
Frequency, therapeutic approaches, and symptoms of dry eye syndrome and allergic conjunctivitis.

**Figure 5 pharmaceutics-15-00470-f005:**
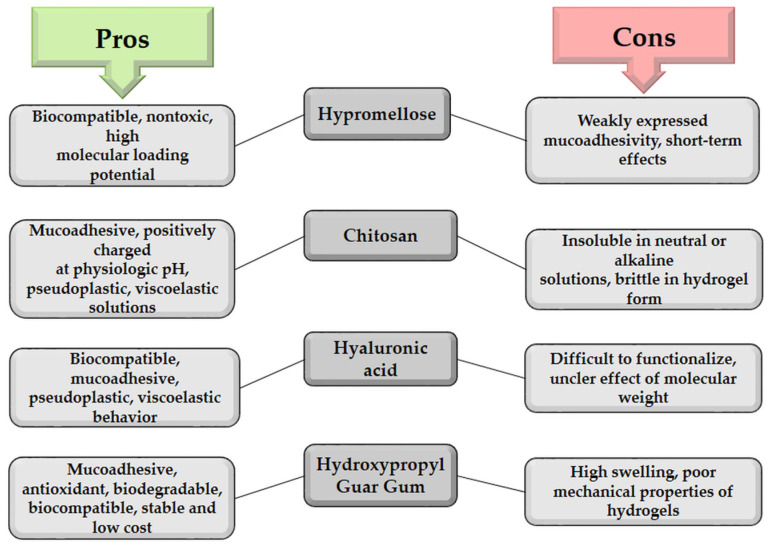
Summary of advantages and disadvantages of some biopolymers in ophthalmic preparations.

**Figure 6 pharmaceutics-15-00470-f006:**
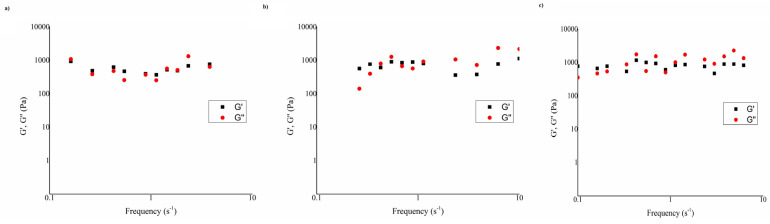
The dependence of the elastic-storage (G′) and viscous-loss (G″) moduli on frequency for polymer-solution–mucin mixtures: (**a**) F2M, (**b**) F7M, (**c**) F8M (*n* = 3, SD values were less than ± 5%). Reprinted with permission from Ref. [101]. Copyright 2019 Elsevier.

**Table 2 pharmaceutics-15-00470-t002:** The basic characteristics of biopolymers in ophthalmic preparations for dry eye syndrome treatment.

Polymer Name	Characteristics	Mucoadhesive Potential ^a^	Role in Artificial Tears	Chemical Structure	Reference
Chitosan	Cationic, non-toxic, biodegradable, biocompatible, soluble in acetic acid	+++	Mucoadhesive polymer	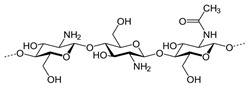	[95]
Hyaluronic acid	Anionic, biocompatible, viscoelastic, mucin-like polymer, slightly soluble in water	+++	Active ingredient/vehicle, mucoadhesive polymer	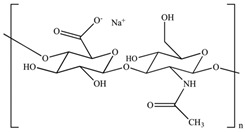	[96]
Hypromellose	Non-ionic, biocompatible, biodegradable material, transparency, water-soluble	+	Active ingredient/tear substitution	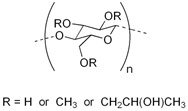	[97]
Xanthan Gum	Anionic, low-toxic, high stability	++	Mucoadhesive polymer, prolonged retention time	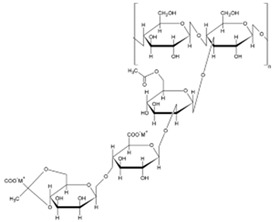	[98]
Hydroxypropyl Guar Gum	Non-ionic, viscosity-increasing polymer, effective at low concentrations, produces clear solutions	++	Gelling agent	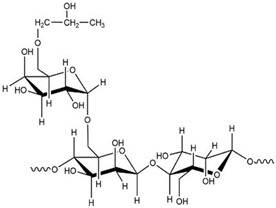	[99]
Gellan Gum	Anionic, mucoadhesive, thixotropic, pseudoplastic	++	Mucoadhesive polymer, prolonged retention time	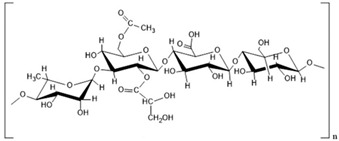	[72]

^a^ Mucoadhesive strenth: +++, strong; ++, medium; + low.

## Data Availability

Not applicable.

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
