# Peer review of "Biopolymers in Mucoadhesive Eye Drops for Treatment of Dry Eye and Allergic Conditions: Application and Perspectives"

_pharmaceutics, 2023, doi:10.3390/pharmaceutics15020470_

Round 1

Reviewer 1 Report

The review article from Račić and Krajišnik, „Biopolymers in Mucoadhesive Eye Drops for Treatment of Dry Eye and Allergic Conditions: Application and Perspectives“, would be interesting for a broader community. Nevertheless, this review leaves mixed feelings. Overall, it covers the main aspects of the topic, such as an introduction to ocular anatomy, mucoadhesion, and dry eye syndrome. Also the usage of polymers in the ophthalmic formulation and how it affects the patient`s behavior. But I find the cartoons too schematic, loosely connected to the text. Additional illustrations would improve the quality of this work. The text itself is poorly written, and I am confused about the main story elements in this review. I suggest rewriting this manuscript.

Let's start with the Abstract section: engaging, with a detailed summary of this work. But is the main idea to describe all available formulations? For me, the main message to take home here is „some polymers are suitable, some are bad“. Thus it very abstractive. I suggest shortening the manuscript and change the arrangement and the flow:
i)    the Introduction section describes the ocular structure and primary eye disorders. The cartoon should be changed, it is unclear what colors or structures shown in Figure 1 mean.
ii)    Then the next Section Mucin and adhesion should focus on how drug vehicles or drugs enter and dissolve in the eye. All aspects of interaction, the discomfort of the patients, reduced or due to patient behavior, insufficient concentration of the drug, etc.
iii)    Lastly, a comparison of biopolymers. There is no chemical formula in this paper.
iv)    Consider rewriting the manuscript. More simplified text with fewer detailes but good reasoning and discussion would improve this review article.
Finally, I like the manuscript but suggest adding more iliustrations or tables where you compare different materials

Author Response

Responses to Reviewer’s comments

Reviewer #1

Reviewer #1 comments:

The review article from Račić and Krajišnik, „Biopolymers in Mucoadhesive Eye Drops for Treatment of Dry Eye and Allergic Conditions: Application and Perspectives“, would be interesting for a broader community. Nevertheless, this review leaves mixed feelings. Overall, it covers the main aspects of the topic, such as an introduction to ocular anatomy, mucoadhesion, and dry eye syndrome. Also the usage of polymers in the ophthalmic formulation and how it affects the patient`s behavior. But I find the cartoons too schematic, loosely connected to the text. Additional illustrations would improve the quality of this work. The text itself is poorly written, and I am confused about the main story elements in this review. I suggest rewriting this manuscript.

Let's start with the Abstract section: engaging, with a detailed summary of this work. But is the main idea to describe all available formulations? For me, the main message to take home here is „some polymers are suitable, some are bad“. Thus it very abstractive. I suggest shortening the manuscript and change the arrangement and the flow:

  1. i) the Introduction section describes the ocular structure and primary eye disorders. The cartoon should be changed, it is unclear what colors or structures shown in Figure 1 mean.

Answer: We sincerely thank the Reviewer #1 for these comments. The sections Introduction, Ocular Anatomy and Physiology and Dry Eye Syndrome and Allergic Conjunctivitis are shortened and rewritten.

The Figure 1 that represents structure of cornea is modified to make the comprehension easier. The corrected Figure 1 is available in the revised manuscript.

In accordance with these changes, the section Introduction has been revised (lines 42-45):

“The objective of this paper is to give an overview of the formulation approaches related to the introduction of mucoadhesive polymers, as functional excipients, into liquid ophthalmic preparations for treatment of dry eye and/or allergic conditions. The review reveals the advantages of biopolymers application in viscous eye drops and the importance of appropriate evaluation of mucoadhesive properties, depending on the type and mechanism of interactions between polymers and mucin within the tear film.”

The section Ocular Anatomy and Physiology has been shortened and some segments are deleted (lines: 47-51, 64-67, 91-94, 101-107, 121-128, 130-133, 139-141, 163-165, 169-173, 229-237, 249-250).

Also, the section Dry Eye Syndrome and Allergic Conjunctivitis has been shortened, lines: 293-295, 298-300, 310-313, 322-332, are deleted.

The section Polymers as Functional Excipients for Improvement of Ocular Availability has been revised, lines: 367-376, 543-564, 842-845 are deleted.

  1. ii) Then the next Section Mucin and adhesion should focus on how drug vehicles or drugs enter and dissolve in the eye. All aspects of interaction, the discomfort of the patients, reduced or due to patient behavior, insufficient concentration of the drug, etc.

Answer: The section 2.5. Mucin and Mucoadhesion has been revised (lines 169-178):

“After the local application of mucoadhesive eye drops, the polymer interacts with the glycoprotein, allowing the preparation formulation to spread over the surface. The therapeutically active substance drug is then able to diffuse into the target area. Formulations with mucoadhesive polymers have advantages such as prolonged residence time in the precorneal tissue, therefore they slow down the elimination of the drug, reduce the frequency of drug administration and improve compliance. In addition, these polymers often have lubricating properties, which additionally affects the patient's comfort. On the other hand, the main disadvantages of this approach for to the improvement of ocular availability are the difficult dosing that depends on the type of mucoadhesive polymer, as well as the patient's mucus production process.”

One additional reference is cited:

[38] Ameeduzzafar; Imam, S.S.; Abbas Bukhari, S.N.; Ahmad, J.; Ali, A. Formulation and optimization of levofloxacin loaded chitosan nanoparticle for ocular delivery: In-vitro characterization, ocular tolerance and antibacterial activity. Int Biol Macromol 2018, 108, 650-659

iii)    Lastly, a comparison of biopolymers. There is no chemical formula in this paper.

Answer: The authors are very grateful for these remarks. One new table with basic characteristics of the biopolymers and their chemical structures is added (pages 11-12):

Table 2. The basic characteristics of the biopolymers in ophthalmic preparations for dry eye syndrome treatment

Also, one schematic illustration about advantages and disadvantages was made (page 15):

Figure 5. Summary of advantages and disadvantages of some biopolymers in ophthalmic preparations

  1. iv) Consider rewriting the manuscript. More simplified text with fewer details but good reasoning and discussion would improve this review article.

Finally, I like the manuscript but suggest adding more illustrations or tables where you compare different materials

Answer: We are grateful for these suggestions. We hope that shortening, rearranging, correcting existing figures and adding additional table and illustration contributed to a better quality of the review. The authors are ready to respond to any further questions and comments.

Reviewer 2 Report

In this study, the authors investigate the formulation approaches related to the introduction of mucoadhesive polymers, as functional excipients, into formulation of liquid ophthalmic preparations for treatment of dry eye and allergic conditions. In addition, authors show that, some polymers, such as chitosan, apart favorable mucoadhesive characteristics and influence on viscosity, also improve the drug permeation by acting through different mechanisms.

So, I recommend its publication after minor modifications.

1.  As this paper can be interesting for the researchers in the field. So, the authors can improve the introduction by expansion the recent developments in this field. The novelty of the present work should be added to the introduction.  

2.  Also, the reference section can strengthen by including some of the new reference papers.

3.  All figures should be enhanced.

4.  The English language of the manuscript should be improved. English language should be polished by a native speaker

Author Response

Responses to Reviewer’s comments

Reviewer #2

Reviewer #2 comments:

In this study, the authors investigate the formulation approaches related to the introduction of mucoadhesive polymers, as functional excipients, into formulation of liquid ophthalmic preparations for treatment of dry eye and allergic conditions. In addition, authors show that, some polymers, such as chitosan, apart favorable mucoadhesive characteristics and influence on viscosity, also improve the drug permeation by acting through different mechanisms.

So, I recommend its publication after minor modifications.

  1. As this paper can be interesting for the researchers in the field. So, the authors can improve the introduction by expansion the recent developments in this field. The novelty of the present work should be added to the introduction.

Answer: The authors are very grateful to the Reviewer #2 for this comment. The section Introduction and the objective of paper has been revised (lines 42-45):

“The objective of this paper is to give an overview of the formulation approaches related to the introduction of mucoadhesive polymers, as functional excipients, into liquid ophthalmic preparations for treatment of dry eye and/or allergic conditions. The review reveals the advantages of biopolymers application in viscous eye drops and the importance of appropriate evaluation of mucoadhesive properties, depending on the type and mechanism of interactions between polymers and mucin within the tear film.”

Also, in accordance with the revised objective, the section Mucoadhesive Biopolymers In the Ophthalmic Formulation is supplemented (lines 456-472):

“During the formulation of ophthalmic preparations, the combinations of polymers can be used that exhibit a synergistic effect on viscosity and rheological behavior or enable more pronounced mucoadhesive properties can be, used. It has been confirmed that the combination of hyaluronic acid and chitosan, but also hyaluronic acid and polysaccharides from tamarind seeds (lat. Tamarindus indica) exhibit a synergistic mucoadhesive effect [119,120]. The presence of gelatin improves the mucoadhesive properties of carrageenan [121] and hyaluronic acid, due to the formation of a soluble polymer-polymer interaction product [122]. The combination of trehalose and hyaluronic acid reduces the desiccation of membrane lipids and proteins and improves the viscoelasticity of the tear film [123]. In an in vitro study on cultured human corneal epithelial cells, the positive effect of combining hyaluronic acid and hydroxypropyl guar gum was confirmed, when compared to the application of the mentioned these polymers individually. The combination of polymers enabled more efficient hydration and lubrication, i.e. reduction of drying over a longer period of time, compared to the individual polymers [124]. Also, there are confirmations in the literature about the positive rheological synergism of xanthan gum and guar gum, which through mutual interaction contribute to longer retention of the preparation at the site of application and better ocular availability.”

As well as the section Concluding Remarks (lines 834-836):

“The combination of two or more polymers significantly contributes to viscosity and lubrication, and allows decreasing the concentration of individual polymers, reducing the possibility of side effects.”

Several additional references are cited:

[123] Pinto-Bonilla, J.C.; del Olmo-Jimeno, A.; Llovet-Osuna, F.; Hernández-Galilea, E. A randomized crossover study comparing trehalose/hyaluronate eyedrops and standard treatment: patient satisfaction in the treatment of dry eye syndrome. Ther Clin Risk Manag 2015, 11, 595.

[124] Rangarajan, R.; Kraybill, B.; Ogundele, A.; Ketelson, H.A. Effects of a hyaluronic acid/hydroxypropyl guar artificial tear  solution on protection, recovery, and lubricity in models of corneal epithelium. J Ocul Pharmacol Ther 2015, 31, 491-497.

[125] Bhowmik, M.; Kumari, P.; Sarkar, G.; Bain, M.K.; Bhowmick, B.; Mollick, M.M.R.; Mondal, D.; Maity, D.; Rana, D.; Bhattacharjee, D. Effect of xanthan gum and guar gum on in situ gelling ophthalmic drug delivery system based on 1124 poloxamer-407. Int J Biol Macromol 2013, 62, 117-123.

  1. Also, the reference section can strengthen by including some of the new reference papers.

Answer: We would like to thank reviewer for this suggestion. In accordance with these remark, some new reference papers have been added in the reference section:

[13] Frutos-Rincon, L.; Gomez-Sanchez, J.A.; Inigo-Portugues, A.; Acosta, M.C.; Gallar, J. An Experimental Model of Neuro- Immune Interactions in the Eye: Corneal Sensory Nerves and Resident Dendritic Cells. Int J Mol Sci 2022, 23.

[14] Downie, L.E.; Bandlitz, S.; Bergmanson, J.P.; Craig, J.P.; Dutta, D.; Maldonado-Codina, C.; Ngo, W.; Siddireddy, J.S.; Wolffsohn, J.S. BCLA CLEAR-Anatomy and physiology of the anterior eye. Cont Lens Anterior Eye 2021, 44, 132-156.

[22] Mofidfar, M.; Abdi, B.; Ahadian, S.; Mostafavi, E.; Desai, T.A.; Abbasi, F.; Sun, Y.; Manche, E.E.; Ta, C.N.; Flowers, C.W. Drug delivery to the anterior segment of the eye: A review of current and future treatment strategies. Int J Pharm 2021, 607.

[24] Rahman, M.M.; Kim, D.H.; Park, C.K.; Kim, Y.H. Experimental Models, Induction Protocols, and Measured Parameters in Dry Eye Disease: Focusing on Practical Implications for Experimental Research. Int J Mol Sci 2021, 22.

[39] Kakkar, S.; Singh, M.; Mohan Karuppayil, S.; Raut, J.S.; Giansanti, F.; Papucci, L.; Schiavone, N.; Nag, T.C.; Gao, N.; Yu, F.X.; et al. Lipo-PEG nano-ocular formulation successfully encapsulates hydrophilic fluconazole and traverses corneal and non-corneal path to reach posterior eye segment. J Drug Target 2021, 29, 631-650.

  1. All figures should be enhanced.

Answer: We highly appreciate the reviewer’s comment regarding the figures. All the figures have been improved in the revised version of the manuscript.

  1. The English language of the manuscript should be improved. English language should be polished by a native speaker

Answer: We regret that there were problems with the English style. The paper has been carefully revised by a professional language editing service to improve the grammar and readability.

Reviewer 3 Report

Dear Authors,

This paper is an interesting work, well documented, and could be a valuable paper in the approached field if the Authors take into consideration the following corrections / suggestions:

1. As a suggestion, the Section 2 “Ocular Anatomy and Physiology” is too expanded. The Authors should reduce this part, especially Subsections 2.1-.2.4.

2. The same observation for the Section 3 “Dry Eye Syndrome and Allergic Conjunctivitis”. Please reduce the Lines 318-333.

3. Please re-numbering the Section “Consideration of Surface Tension” as 5.2 instead 5.1.

4. As a general observation: the Authors excessively cite the references represented by the books. For example, references [1, [3], [13], [14] …..

As a suggestion, for example the Authors could cite articles to underline the low availability for lypophilic and hydrophilic molecules (See Lines 216-219).

5. Please replace utilized” with used.

6. Please rephrase the Lines 38-42.

7. Other minor English revision and spelling corrections are required.

Author Response

Responses to Reviewer’s comments

Reviewer #3

Reviewer #3 comments:

This paper is an interesting work, well documented, and could be a valuable paper in the approached field if the Authors take into consideration the following corrections / suggestions:

  1. As a suggestion, the Section 2 “Ocular Anatomy and Physiology” is too expanded. The Authors should reduce this part, especially Subsections 2.1-.2.4.

Answer: We sincerely thank the Reviewer #3 for all the comments.

The section Ocular Anatomy and Physiology has been shortened and some segments were deleted (lines: 47-51, 64-67, 91-94, 101-107, 121-128, 130-133, 139-141, 163-165, 169-173, 229-237, 249-250).

  1. The same observation for the Section 3 “Dry Eye Syndrome and Allergic Conjunctivitis”. Please reduce the Lines 318-333.

Answer: The Section Dry Eye Syndrome and Allergic Conjunctivitis has been revised and lines 293-295, 298-300, 310-313, 322-332, were deleted.

  1. Please re-numbering the Section “Consideration of Surface Tension” as 5.2 instead 5.1.

Answer: The authors are grateful for this remark. The correction of section numbering has been made.

  1. As a general observation: the Authors excessively cite the references represented by the books. For example, references [1, [3], [13], [14] …..

As a suggestion, for example the Authors could cite articles to underline the low availability for lypophilic and hydrophilic molecules (See Lines 216-219).

Answer: We highly appreciate the reviewer’s suggestion regarding of the references. According to Reviewer’s comment, relevant recent papers are cited instead of books. Also, for mentioned section which refers to the low availability for lipophilic and hydrophilic molecules (lines 216-219), one new reference is added:

[36] Zhang, W.; Prausnitz, M.R.; Edwards, A. Model of transient drug diffusion across cornea. Journal of Controlled Release 2004, 927 99, 241-258.

  1. Please replace “utilized” with used.

Answer: The replacement has been made.

  1. Please rephrase the Lines 38-42.

Answer: This section has been revised to:

“To improve the therapeutic efficiency of existing drug formulations, various multifunctional biopolymers are being used to increase viscosity of the tear film, exhibit bioadhesive properties, and/or to increase corneal permeability. Additionally, these substances are also used as active ingredients in artificial tear preparations due to their lubricating and moisturizing effect.”

  1. Other minor English revision and spelling corrections are required.

Answer: We regret that there were problems with the English style. The paper has been carefully revised by a professional language editing service to improve the grammar and readability.

Round 2

Reviewer 1 Report

Regarding revision of the manuscript "Biopolymers in mucoadhesive eye drops for treatment of dry eye and allergic conditions: application and perspectives" (manuscript ID: pharmaceutics-2152289).
The authors improved the manuscript a lot, and as it is valuable for the broader scientific community, I think it should be published in the journal.

Nevertheless, currrent version needs some proofreading. Just several examples: 1) reference numbers in the text appear chaotically, ex. 1,2 next 4,5 - number 3 is missing, after [8] there is [11]; 2) In Figure 6 both axes should be indicated as logarithmic; 3) line 869 the "." is missing; 4) line 461 "strength" etc.

Reviewer 3 Report

Dear Authors,

All suggested revision / corrections were made.